# The estrogen effect; clinical and histopathological evidence of dichotomous influences in dogs with spontaneous mammary carcinomas

**Karin U. Sorenmo**[1,2]*, **Amy C. Durham**[2,3], **Enrico Radaelli**[3], **Veronica Kristiansen**[4¤a], **Laura Peña**[5], **Michael H. Goldschmidt**[3¤b], **Darko Stefanovski**[6]

**1** Department of Biomedical Sciences, School of Veterinary Medicine, University of Pennsylvania, Philadelphia, Pennsylvania, United States of America, **2** Penn Vet Cancer Center, University of Pennsylvania, Philadelphia, Pennsylvania, United States of America, **3** Department of Pathobiology, School of Veterinary Medicine, University of Pennsylvania, Philadelphia, Pennsylvania, United States of America, **4** Department of Companion Animal Clinical Sciences, Veterinary Medicine and Biosciences, Norwegian University of Life Sciences, Oslo, Norway, **5** Department of Animal Medicine, Surgery and Pathology, Veterinary School, Complutense University of Madrid, Madrid, Spain, **6** Department of Clinical Studies, School of Veterinary Medicine, University of Pennsylvania - New Bolton Center, Kennett Square, Pennsylvania, United States of America

¤a Current address: Evidensia Animal Hospital, Oslo, Norway
¤b Current address: Hilton Head Island, South Carolina, United States of America
* karins@vet.upenn.edu

**Data Availability Statement:** Clinical data, outcomes, histology, hormone receptors, and serum hormone results are within the paper and its

## Abstract

The purpose of this study was to investigate the associations and explore the relationships between hormonal factors (serum estrogen, estrogen receptors and ovariohysterectomy) and other clinical/histological prognostic factors and their impact on outcome in dogs with mammary carcinomas. Data from two separate prospective studies on dogs with spontaneous mammary carcinomas were used for this research. All dogs underwent standardized diagnostic testing, staging, surgery and follow-up examinations. Serum estrogen was analyzed by competitive enzyme immunoassay or radioimmunoassay, and tumor estrogen receptor (ER) expression was analyzed by immunohistochemistry. A total of 159 dogs were included; 130 were spayed and 29 remained. High serum estrogen was associated with an overall longer time to metastasis (p = 0.021). When stratifying based on spay group, the effect was only significant in spayed dogs, (p = 0.019). Positive tumor ER expression was also associated with a longer time to metastasis (p = 0.025), but similar to above, only in dogs that were spayed (p = 0.049). Further subgroup analysis revealed that high serum estrogen was significantly associated with improved survival in dogs with ER positive tumors, but only in spayed dogs (p = 0.0052). Interestingly, the effect of spaying was the opposite in dogs with ER negative tumors; here, intact dogs with high serum estrogen but ER negative tumors had a significantly longer time to metastasis (p = 0.036). Low serum estrogen was associated with increased risk for the development of non-mammary tumors in the post-operative period (p = 0.012). These results highlight the dual effect of estrogen in cancer: Estrogen acts as a pro-carcinogen in ER positive mammary tumors, but a may have a protective effect in ER negative tumors, potentially via non-receptor mechanisms. The

Supporting Information files. The original data set has been uploaded to figshare and is accessible using the following doi: 10.6084/m9.figshare.9806033.

**Funding:** This work was supported through a grant from The Petco Foundation and the Blue Buffalo Foundation (no grant number available) to KS to support the clinical work and research developed from the PennVet Shelter Canine mammary Tumor program. The founders had no role in study design, data collection and analysis, decision to publish, or preparation of the manuscript.

**Competing interests:** The authors have declared that no competing interests exist.

latter is supported by the decreased risk for non-mammary tumors in dogs with high serum estrogen, and explains the increased incidence of certain non-mammary tumors in in dogs spayed at an early age.

## Introduction

Estrogen has been recognized as a major driver of breast carcinogenesis in both women and dogs. In both species, the breast cancer risk is directly correlated to the duration of exposure of mammary tissue to bioavailable estrogens [1–7]. These epidemiological observations are further supported by studies confirming higher serum estrogen levels in women and dogs with breast cancer than age-matched controls without breast cancer. Specifically, high serum estrogen is associated with increased risk of breast cancer in both pre and post-menopausal women and also reported to be associated with increased risk of relapse in post-menopausal women [8–10]. Similarly, dogs with non-inflammatory mammary carcinomas had significantly higher serum estrogen than dogs with no mammary tumors or dogs with inflammatory mammary carcinomas [11]. The principal mechanism by which estrogen initiates and drives breast cancer is via the estrogen receptor, but receptor independent effects has also been reported [12–17]. The biological and molecular consequences of estrogen binding to the nuclear receptors in breast epithelial cells have been examined in numerous studies and provide mechanistic explanations for the pro-breast carcinogenic effects of estrogen. By entering the cells and binding the nuclear receptor, estrogen initiates a sequence of molecular events resulting in altered transcription of estrogen responsive genes (ERGs), thus causing increased expression of positive proliferation regulators and down-regulation of anti-proliferative and pro-apoptotic genes [18–20]. The net effect is increased cell division facilitating continued growth and promoting additional spontaneous mutations [12, 15, 18, 21]. Collectively, this data provides compelling corroboration for the current view that estrogen has deleterious effects in patients with breast cancer. As a result, much of the therapeutic strategies in human breast cancer revolve around averting the interaction between ligand and receptor through various estrogen receptor inhibitors, aromatase inhibitors, or oophorectomy/ovariohysterectomy [22–25]. The decision to use hormonal therapy in women with breast cancer is based on results from tumor receptor analysis via IHC; serum estrogen level analysis is typically not performed because estrogen levels change significantly with frequent spikes through the menstrual cycle and therefore are difficult to use in treatment decisions [26].

The estrus cycle in dogs differs significantly from the menstrual cycle in women. Depending on the size and breed, dogs generally have 2 estrus cycles per year with prolonged periods of diestrus and anestrus between each proestrus/estrus phase, both of which are of relatively short duration (1–3 weeks) [27]. Consequently, the fluctuations in serum estrogen are less pronounced with fewer estrogen spikes. Therefore, serum estrogen levels in dogs with mammary tumors may reflect a more accurate representation of the average serum estrogen, and less likely to be skewed due to spikes during the follicular phase. Thus, studying estrogen levels in dogs with mammary carcinomas may provide clarity and new insight into the effect of estrogen on tumor gene expression and tumor biology. Results from our recent study on the effect of ovarian hormonal ablation (ovariohysterectomy, OHE) as an adjuvant to mammary tumor removal showed that, contrary to expectations, dogs with high serum estrogen had significantly longer disease-free survival and overall survival than dogs with low serum estrogen [28]. Notably, this effect was observed only in dogs that were randomized to undergo OHE

concurrent with tumor removal. Hence, it was proposed that serum estrogen might be used as a surrogate marker for estrogen dependence and replace the use of immunohistochemistry (IHC) for estrogen receptor analysis. Subsequent unpublished subgroup analysis revealed that the effect was more complex and most pronounced in dogs with histologically aggressive tumors (predominantly grade 3 tumors), suggesting a potentially ER-independent mediated effect. Based on these observations, a more comprehensive study of a larger population of dogs with mammary carcinomas was undertaken.

The purpose of this study was to explore the effect of serum estrogen in canine mammary carcinomas and its relationship with OHE status, bioscore group, tumor grade, tumor stage and tumor ER immunohistochemistry. Outcomes included the development of metastatic disease, de novo mammary tumors, other non-mammary tumors, and overall survival time. Based on our pilot study results, our first hypothesis was that the benefit from high serum estrogen would be strongest in dogs with histologically aggressive tumors, ER negative tumors, and high bioscores—all indicators of a worse prognosis. Secondly, we hypothesized that if the survival benefit seen in dogs with high serum estrogen was mediated via non-estrogen receptor mechanisms, these dogs would also be less likely to develop other non-mammary tumors in their post-operative period. Here we provide further evidence that serum estrogen is a major player in canine mammary carcinomas, confirming our hypothesis that high serum estrogen significantly prolongs time to metastasis and survival in dogs with high risk tumors and/or ER-negative tumors, but also predicts improved survival in dogs with ER-positive tumors if they undergo OHE. Additionally, the lower incidence of certain non-mammary malignancies in the follow-up period in dogs with high serum estrogen supports the theory that this effect may in part be mediated via non-receptor mechanisms and may explain the epidemiological reports regarding an increased risk of malignancies in dogs that undergo OHE at an early age.

## Materials and methods

Clinical data: Information from dogs with mammary carcinomas of various subtypes enrolled in two separate prospective studies were combined and used for this research. Specifically, 60 dogs were enrolled in a prospective randomized trial of the effect of OHE in canine mammary carcinoma (Morris Animal Foundation study, MAF). The other group consisted of 99 dogs enrolled in the PennVet Shelter Canine Mammary Tumor Program (PVSCMTP). All intact dogs underwent OHE as part of their tumor removal in this program. Both studies were approved by their respective Institutional Animal Care and Use Committee at the School of Veterinary Sciences, Oslo, Norway and the University of Pennsylvania, Philadelphia, PA. A written consent was obtained from all owners and shelter representatives prior to enrollment. The pre-surgical staging and follow-up examinations were standardized for both groups and described in detail in recent publications [28, 29]. All dogs underwent surgical resection of all measurable tumors as per the surgical technique deemed necessary to achieve complete margins. Dogs in the MAF study were randomized to undergo concurrent OHE or remain intact. All intact shelter dogs underwent OHE at the first time of tumor removal. Owners or fosters of dogs with new mammary tumors were advised to have their dog's tumor resected, however, no financial incentive was provided for this in the MAF study. Dogs in the PVSCMTP had free mammary care for the remainder of their life and dogs with tumors larger than 1 cm were recommended to have another surgery. Metastases were classified as primary metastasis (M1, metastasis from the current mammary tumor) or secondary metastasis (M2, metastasis from later mammary tumors) according to the scheme described previously [29]. Necropsies were requested on all dogs. Cause of death was classified as mammary tumor related versus other causes. In addition, the incidence of other non-mammary tumors (NMTs) and types of NMTs

was also collected from the medical records and the necropsy reports when available. None of the MAF dogs could pursue chemotherapy until distant metastasis was noted. The PVSCMTP dogs could pursue chemotherapy after surgery, but the cost for such treatments were not covered by the program. Fosters or new owners of dogs with tumors that were perceived to be associated with significant risk for metastasis were advised that chemotherapy might be beneficial.

Histopathology and bioscoring: All tumors and associated lymph nodes (when available) were fixed in 10% buffered formalin, processed according to standard procedures and stained with H&E for histopathological evaluation. The tumors were graded and classified according to histological subtypes. In addition, the presence of vascular invasion and completeness of surgical margins were noted (ACD, LP, MHG) [30, 31]. All dogs were assigned a prognostic bioscore based on the refined flexible scoring system (RFS) where tumor grade, subtype, size and WHO stage (tumor size and lymph node status) were included and scored as described in our previous study on prognostication in canine mammary tumors [29].

Estrogen Receptor analysis was performed following standardized criteria [32]: The ER IHC for the MAF dogs was performed and interpreted at the Laboratory of Pathology, Veterinary Clinical Hospital at the Veterinary School of the Complutense University of Madrid, Spain (LP). Here the clone 1D5 (M7047, Dako, Glostrup, Denmark) was used as described previously [27]. IHC for the PVSCMTP dogs' tumors was performed and scored at the Penn Vet diagnostic laboratory (ACD). As the antibody used for the MAF dogs was unavailable, another ERα (Agilent/DAKO #IR084) was validated for canine mammary tumors via the following methods: 5 μm thick paraffin sections were mounted on ProbeOn™ slides (Thermo Fisher Scientific). The immunostaining procedure was performed using a Leica BOND RXm automated platform combined with the Bond Polymer Refine Detection kit (Leica #DS9800). Briefly, after dewaxing and rehydration, sections were pretreated with the epitope retrieval BOND ER2 high pH buffer (Leica #AR9640) for 20 minutes at 98˚C. Endogenous peroxidase was inactivated with 3% H2O2 for 10 minutes at RT. Nonspecific tissue-antibody interactions were blocked with Leica PowerVision IHC/ISH Super Blocking solution (PV6122) for 30 minutes at RT. The same blocking solution also served as diluent for the primary antibody. Rabbit monoclonal primary antibodies against ERα (Agilent/DAKO #IR084) was used at a concentration of 1/30 and incubated on the sections for 45 minutes at RT. A biotin-free polymeric IHC detection system consisting of HRP conjugated anti-rabbit IgG was then applied for 25 minutes at RT. Immunoreactivity was revealed with the diaminobenzidine (DAB) chromogen reaction. Slides were finally counterstained in hematoxylin, dehydrated in an ethanol series, cleared in xylene, and permanently mounted with a resinous mounting medium (Thermo Scientific ClearVueTM coverslipper). Sections of canine uterus were included as positive controls. Negative controls were obtained either by omission of the ERα antibody or replacement with an irrelevant isotype-matched rabbit monoclonal antibody.

The Allred scoring system was used to score the ER immunohistochemical (IHC) staining in both groups. In this method both the percentage of stained cells (PS): none = 0, <1% = 1, 1–10% = 2, 10–33% = 3, 33–66% = 4, 66–100% = 5) and the staining intensity (IS): absent = 0, weak = 1, moderate = 2, and strong = 3) are measured and summarized as a total Allred score (TS). The total score (PS + IS) was equal to 0 or ranged from 2 to 8. Based on the previous published work with the MAF dogs, the established cut-off for positivity of $\geq 3$ based on Allred scoring was used for all dogs [28]. The results from the MAF dogs have been published previously [28]. The PVSCMTP cases were scored by a single board-certified pathologist (ACD) and data from these two groups combined. Serum estrogen analysis: Serum samples were collected from all dogs immediately prior to surgery and stored in -80 F until analysis. Two different methods in two different laboratories were used. The MAF study dogs' serum samples

were analyzed using competitive enzyme immunoassay (EIA) at the Laboratory of Physiology of the Veterinary School of the Complutense University of Madrid, Spain [11, 28] and the PVSCMT samples were analyzed by radioimmuno assay (RIA) at the Clinical Endocrinology Service, Department of Biomedical & Diagnostic Sciences, College of Veterinary Medicine of the University of Tennessee, which provides an established commercial hormone analysis service. The median value of estradiol (E2) in the MAF group was 35 pg/ml and the median value in the PVSCMTP group was 39 pg/ml. There was no significant difference in serum E2 analyzed by EIA and RIA, (p = 0.59 Mann-Whitney). The MAF dogs were classified as high or low group based on 35 pg/ml cut-off while the PVSCMTP dogs were classified as high vs low based on 39 pg/ml as the cut-off value.

## Statistics, endpoints and variables

Descriptive statistics for continuous variables were reported as median ± IQR. Frequency counts and percentages were used for categorical variables. In order to provide a complete understanding of the biological effects of estrogen and OHE on all mammary tumor related failures and survival, three endpoints were analyzed: (1) time to primary metastasis (TTM); (2) disease free survival (DFS: including local recurrence, new mammary tumors and any mammary tumor metastasis); and (3) overall survival (OS). Dogs that were lost to follow-up or were alive or dead with no evidence of metastasis or local recurrence/new tumors (LRNT) were censored at the date of last known status in the TTM and DFS analysis. Dogs that were lost or alive were censored in the survival analysis. The Kaplan Mayer product limit method and the log rank test were used to compare outcomes across the selected variables.

Two different statistical approaches were undertaken: First, a standard exploratory univariate analysis followed by inference multivariable approach where the dogs were treated as one group; and second, a more individualized approach where dogs were stratified according to prognostic factors (tumor grade, stage, and bioscore). The impact of OHE, serum E2, and tumor ER expression were analyzed in the various subgroups to identify variations in outcomes that might be lost in the multivariable analysis. In addition, to further evaluate the interaction between the 3 hormonal factors: serum E2 (high versus low), tumor receptor status (ER positive versus negative) and OHE status (spayed versus intact), Cox regression analysis was performed, both using primary metastasis and overall survival as outcomes. Variables tested in univariate analysis included serum estrogen (categorical: high versus low and continuous), OHE (spayed vs intact), ER IHC (positive vs negative, based on Allred scoring), tumor grade, tumor stage, RFS bioscore (>3 or ≤ 3). In addition, standard proportional hazard (Cox regression) multivariable analysis was performed using factors that showed association with the outcome with P-values <0.2 in the exploratory univariate analysis. The final models for both TTM and OS were created through stepwise Cox regression analysis. All analyses were performed using Stata 15MP, StataCorp, State College TX, and MedCalc for Windows, version 18.6 (MedCalc Software, Ostend, Belgium). Two-sided tests of hypotheses and a p-value < 0.05 was used as the criterion for statistical significance.

In order to further identify where the estrogen effect appeared the most beneficial and the role of OHE in potentially mediating this effect, the dogs were stratified into subgroups. Here, we compared outcomes (TTM and OS) in dogs with high vs low serum E2 in dogs with grade 1, 2, or 3 tumors, stage 4 disease, ER negative vs positive disease, and dogs with high versus low RFS bio-scores in both the OHE group and the intact group. In addition, the effect of OHE (spayed versus intact) in ER positive tumors was evaluated in a sub-group of dogs with high risk tumors based on bio-score, RFS >3 and dogs with stage 4 disease (Lymph Node positive tumors).

In addition, we also evaluated the association between serum E2 and tumor grade, ER expression, and the effect on the development of other NMTs in the spayed group. Here, E2 was analyzed both as a categorical and continuous variable where E2 was compared across groups based on histological grades (1, 2, 3), ER IHC (positive vs negative), and dogs that developed other NMTs, including specific histologic types of other NMTs, and dogs that did not fail due to mammary tumor metastasis or NMTs. The chi square test and linear regression were used when comparing categorical and continuous data, respectively. In order to further study the association between E2 and specific histological types of NMTs, we performed pair-wise comparison between dogs grouped according to NMT types and a control group consisting of dogs with no metastasis from their mammary tumors or evidence of NMT during their follow-up period.

## Results

At the time of study completion, 97 of the 159 cases had died, 68 of which had post-mortem examinations. Forty-one of the 130 (31.5%) of the dogs in the OHE group developed local recurrence or new tumors (LRNT), and 27 (20.7%) have developed primary metastasis. In the 29 intact dogs, 17 (58.6%) and 9 (31%) developed LRNT and metastasis, respectively. The incidence of LRNT was significantly lower in dogs that were spayed than dogs that remained intact (p = 0.0097). The difference in primary metastasis between the 2 groups did not reach statistical significance (p = 0.23). Twenty-eight of the 130 dogs (21.5%) in the spayed group developed other NMTs. Three of the twenty-nine dogs (10.3%) in the intact group developed other NMTs. There was no difference in NMT development between the intact and the spayed dogs (p = 0.204). Two dogs received adjuvant chemotherapy, based on a combination of doxorubicin, cyclophosphamide and fluorouracil. One of these dogs had a stage 3, grade 3 carcinoma and developed distant metastasis 407 days after surgery. The other dog had a grade 1, stage 4 mammary carcinoma. This dog was free from disease 944 days after surgery. The standard clinical and histological prognostic factors and their associated statistical significance for all dogs without including the impact of the hormonal factors (serum estrogen, tumor ER and OHE status) is provided in S1 Table.

Serum E2 samples were available from 146 dogs for analysis, the median E2 for the whole group was 38.2; high and low serum E2 categories each had 73 (50%) dogs. ER IHC was performed on 123 cases, of these 74 (60.2%) were positive and 49 (39.8%) were negative based on Allred-score ≥3.

### Univariate and multivariable analysis

Univariate analysis: Serum E2 (categorical and continuous), ER IHC, RFS category, stage, and grade were associated with time to primary metastasis (TTM) and overall survival (OS) with a P<0.2 and included in the multivariable model (Table 1). Multivariable analysis: Both tumor grade (p<0.000) and stage (p = 0.019) remained significant for TTM, but serum estrogen did not (p = 0.067). For overall survival, serum E2 categorical (p = 0.02), stage (0.041) and grade (p = 0.001) remained significant in the multivariable analysis (Cox regression).

Notably, when comparing outcomes between groups with log-rank testing, dogs with high serum E2 regardless of OHE status had significantly longer TTM (medians not reached, p = 0.021, Fig 1), and OS (743 in low E2 versus 1051 days in high E2, p = 0.026), confirming the overall impact of high serum E2 on outcome in dogs with mammary carcinomas. Stratification based on OHE status, revealed that this effect was only significant in spayed dogs (p = 0.019). The difference according to E2 category, however, did not reach significance for the DFS (557 in low E2 vs 941 in high E2, p = 0.059). Overall, dogs with ER positive tumors

**Table 1. Cox regression analysis of factors associated with TTM and OS.**

| Variable | Univariate TTM | | Univariate OS | |
|---|---|---|---|---|
| | P-value | HR | P-value | HR |
| E2 Categorical | 0.025 | 0.4326037 | 0.026 | 0.6162954 |
| Continuous | 0.033 | 0.9801034 | 0.036 | 0.9889612 |
| ER IHC | 0.028 | 0.4638845 | 0.126 | 0.7019999 |
| RFS category | <0.000 | 9.248936 | <0.000 | 3.250886 |
| Grade | <0.000 | 12.97099 | <0.000 | 3.465578 |
| Stage | 0.001 | 6.158832 | 0.037 | 1.899516 |

Results from univariate Cox regression analysis for TTM and OS with the associated p-values and respective Hazard Ratio (HR) for the variables included in the multivariable analysis.

regardless of OHE or E2 category also have a significantly longer TTM than dogs with ER negative tumors (TTM not reached in ER positive versus 1294 days in the ER negative cases, p = 0.025, Fig 2). Similar to above, stratification based on OHE status, found that this significance was only observed in the spayed dogs (p = 0.049). ER expression status was not related to differences in DFS and OS (p = 0.29 and 0.126 respectively). Lastly, OHE, regardless of E2 category and ER status, was associated with significantly longer DFS; the median DFS was 422 days for intact dogs and 754 days for spayed dogs (p = 0.026, Fig 3). OHE did not significantly influence TTM (p = 0.411) or OS (p = 0.227).

**Interaction between OHE, ER IHC and serum E2 and results from subgroups.** Based on Cox regression analysis on the effect and interaction between serum E2, OHE status and tumor ER IHC, dogs that remained intact with low serum E2 and ER negative tumors had a significantly increased risk of distant metastasis (p<0.05) compared to all other groups (Table 2). Similarly, intact dogs with low serum E2/ER negative tumors had a significantly increased risk of death (p<0.01) compared to intact dogs with high serum E2/ER negative tumors and spayed dogs with high serum E2/ER positive tumors (Table 3). The effects of E2

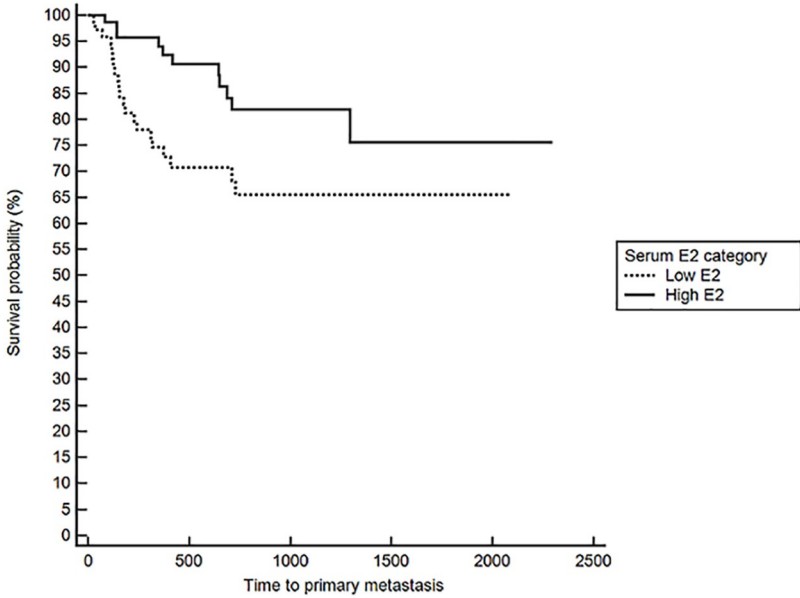

**Fig 1. Time to primary metastasis (M1) according to serum E2 category (low vs high), log rank, p = 0.021.**

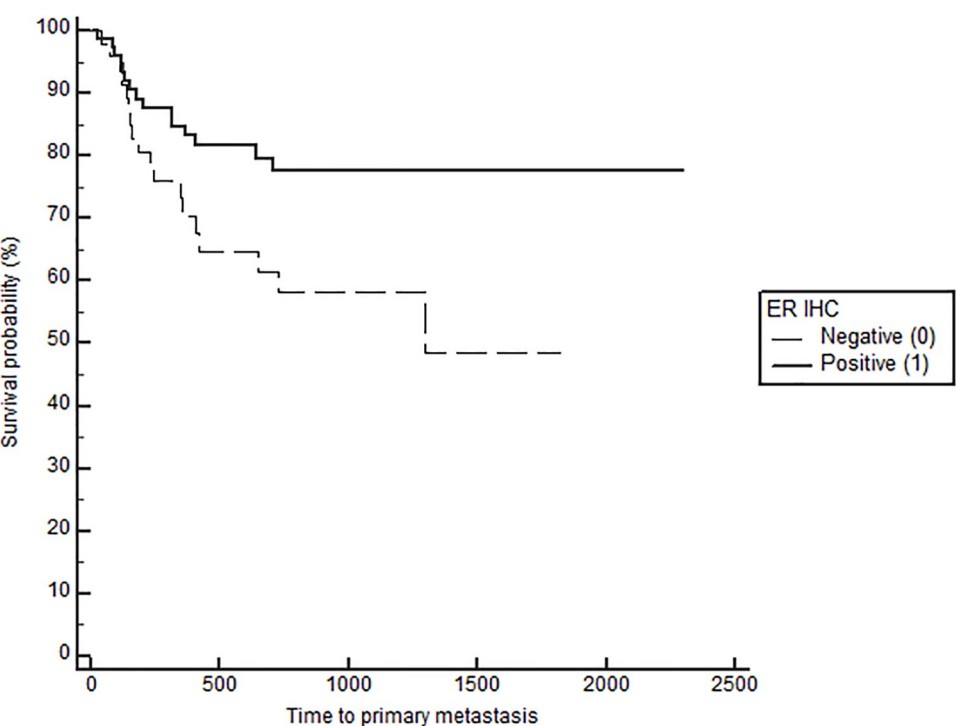

**Fig 2. Time to primary metastasis (M1) according to tumor IHC ER (positive vs negative), log rank, p = 0.025.**

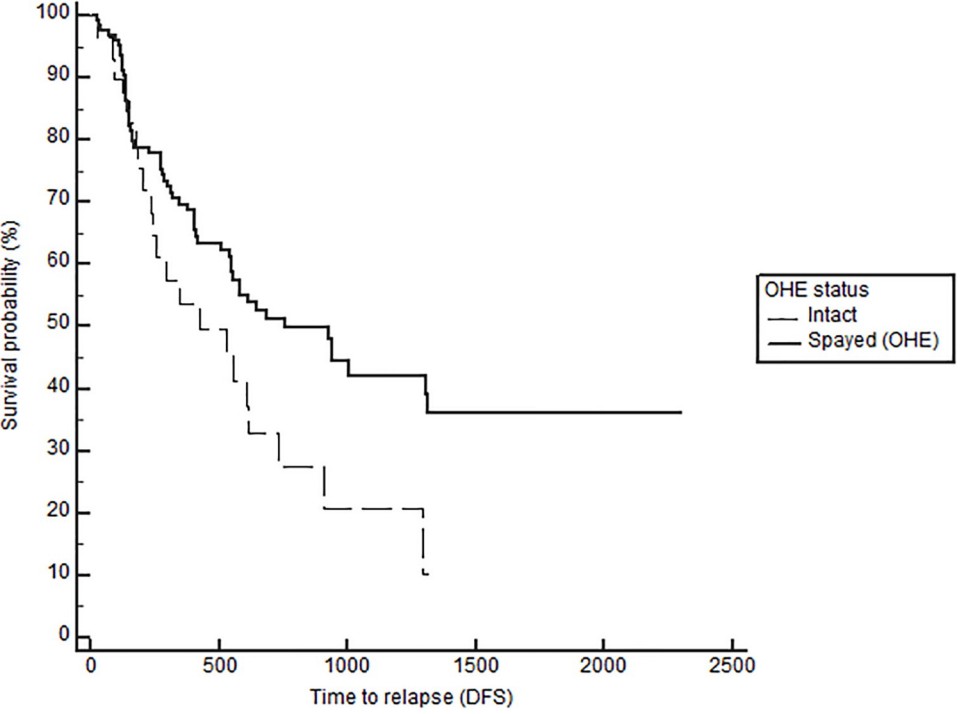

**Fig 3. Disease-free survival (DFS) according to OHE status (spayed vs intact), log rank p = 0.026.**

**Table 2. Interaction of hormonal factors and risk for metastasis.**

| Combined variables | | | Hazard Ratio (HR) | P-value | 95% confidence interval |
|---|---|---|---|---|---|
| E2 | OHE | ER IHC | | | |
| low | intact | ER positive | .0945567 | 0.034 | .0107451–.8320994 |
| low | spayed | ER negative | .3657467 | 0.024 | .1524483–.8774823 |
| low | spayed | ER positive | .3252589 | 0.014 | .1329538–.7957152 |
| high | intact | ER negative | .2067442 | 0.031 | .0495342–.8629015 |
| high | intact | ER positive | .1701368 | 0.021 | .0377617–.7665571 |
| high | spayed | ER negative | .2536103 | 0.004 | .0993608–.6473196 |
| high | spayed | ER positive | .0466294 | 0.000 | .0114101–.1905595 |

Interaction of OHE (spay status) status, serum E2, and tumor ER expression in dogs with mammary carcinomas, and effect on risk for metastasis, compared to control group represented by intact dogs with low E2 and ER negative tumors, Cox regression analysis.

and OHE on TTM and OS in the subgroups based on grade, stage, RFS bioscore, and ER IHC, are presented in Table 4. Neither serum E2 nor OHE significantly affected the outcome in dogs with low-grade tumors. The outcomes are generally good, and the majority of dogs do well long term; median TTM was not reached in these subgroups. As expected, the outcome changes with higher grades, bioscores (RFS) and stage (stage 4). Importantly, this is also where the effect of E2 becomes apparent. The benefit of high E2 is particularly obvious in dogs with stage 4 disease, with a significant difference in TTM and OS both in the spayed and the intact groups (Figs 4 and 5). The benefit associated with high E2 was not statistically significant in dogs with grade 3 tumors when analyzing the data according to OHE group. However, when eliminating the effect of OHE and combining all dogs with grade 3 tumors, the positive impact of high E2 becomes evident: TTM was 184 days in the low E2 group and TTM was not reached in the high E2 group (p = 0.03). Interestingly, dogs with ER positive tumors and high E2 had significantly longer TTM (p = 0.0052) and OS (p = 0.029) than dogs with low E2, but only if they were spayed (Table 4, Fig 6). The opposite was the case in dogs with ER negative tumors; here dogs with high E2 had significantly longer TTM (p = 0.036) and OS (p = 0.036) if they remained intact, uncovering a potentially deleterious effect of removing the ovaries and decreasing the E2 in this subset of dogs (Table 4, Fig 7). Lastly, the subgroup analyses confirmed a benefit of OHE in dogs with high risk tumors according to bioscores (RFS>3) and ER positive tumors, (p = 0.0042 for TTM and p = 0.005 for OS, Table 4, Fig 8). Overall survival was significantly improved in spayed dogs compared to intact dogs with LN metastasis and ER

**Table 3. Interaction of hormonal factors and effect on survival.**

| Combined variables | | | Hazard Ratio (HR) | P-value | 95% confidence interval |
|---|---|---|---|---|---|
| E2 | OHE | ER IHC | | | |
| low | intact | ER positive | .4446109 | 0.143 | .1502207–1.315923 |
| low | spayed | ER negative | .3908093 | 0.111 | .1230857–1.240858 |
| low | spayed | ER positive | .3576298 | 0.061 | .1218319–1.0498 |
| high | intact | ER negative | .0980953 | 0.003 | .0210917–.4562317 |
| high | intact | ER positive | .3949171 | 0.106 | .1278342–1.220014 |
| high | spayed | ER negative | .4364435 | 0.112 | .156984–1.213391 |
| high | spayed | ER positive | .1426007 | 0.000 | .0478878–.4246382 |

Interaction of OHE status, serum E2, and tumor ER expression in dogs with mammary carcinomas, and effect on survival, compared to control group represented by intact dogs with low E2 and ER negative tumors, Cox regression analysis.

**Table 4. Combined clinical/histological and hormonal variables and effect on outcome.**

| Combined Variables | Time to primary metastasis | | | | | Overall survival | | | | |
|---|---|---|---|---|---|---|---|---|---|---|
| | TTM | | | | | OS | | | | |
| | n | spayed | n | Intact | P-value | n | spayed | n | Intact | P-value |
| Grade 1 | | | | | | | | | | |
| E2 high | 39 | NR | 8 | NR | P = 0.88 | 39 | 1183 | 8 | 1051 | P = 0.99 |
| E2 low | 28 | NR | 6 | NR | P = 0.38 | 28 | 1306 | 6 | 758 | P = 0.21 |
| P-value | P = 0.77 | | P = 0.35 | | | P = 0.87 | | P = 0.07 | | |
| Grade 2 | | | | | | | | | | |
| E2 high | 15 | NR | 3 | 1294 | P = 0.41 | 15 | 818 | 3 | 630 | P = 0.32 |
| E2 low | 22 | NR | 4 | NR | P = 0.93 | 22 | 730 | 4 | 150 | P = 0.72 |
| P-value | P = 0.32 | | P = 0.92 | | | P = 0.10 | | P = 0.98 | | |
| Grade 3 | | | | | | | | | | |
| E2 high | 5 | 348 | 3 | NR | P = 0.89 | 5 | 348 | 3 | NR | P = 0.42 |
| E2 low | 10 | 149 | 3 | 240 | P = 0.25 | 10 | 182 | 3 | 240 | P = 0.25 |
| P-value | P = 0.083 | | P = 0.36 | | P = 0.13 | | | P = 0.36 | | |
| Stage 4 | | | | | | | | | | |
| E2 high | 10 | NR | 3 | NR | P = 0.38 | 10 | 926 | 3 | 1459 | P = 0.45 |
| E2 low | 8 | 153 | 2 | 26 | P = 0.99 | 8 | 205 | 2 | 26 | P = 0.63 |
| P-value | **P = 0.047** | | **P = 0.039** | | | **P = 0.049** | | **P = 0.039** | | |
| ER positive | | | | | | | | | | |
| E2 high | 31 | NR | 9 | NR | P = 0.13 | 31 | 1442 | 9 | 630 | **P = 0.033** |
| E2 low | 20 | NR | 8 | NR | P = 0.29 | 20 | 833 | 8 | 758 | P = 0.599 |
| P-value | **P = 0.0052** | | P = 0.66 | | | **P = 0.029** | | P = 0.72 | | |
| ER negative | | | | | | | | | | |
| E2 high | 18 | NR | 3 | 1294 | P = 0.53 | 18 | 807 | 3 | 1590 | **P = 0.03** |
| E2 low | 22 | NR | 3 | 240 | P = 0.16 | 22 | 408 | 3 | 240 | P = 0.4 |
| P-value | P = 0.49 | | **P = 0.036** | | | P = 0.89 | | **P = 0.036** | | |
| ER positive | | | | | | | | | | |
| RFS < 3 | 50 | NR | 16 | NR | P = 0.94 | 50 | 1361 | 16 | 758 | P = 0.06 |
| RFS > 3 | 5 | 149 | 3 | 84 | **P = 0.0042** | 5 | 317 | 3 | 109 | **P = 0.005** |
| P-value | **P<0.0001** | | **P<0.0001** | | | **P = 0.0009** | | **P<0.0001** | | |
| ER negative | | | | | | | | | | |
| RFS < 3 | 29 | NR | 1 | 1254 | P: NC* | 29 | 807 | 1 | 1590 | P: NC* |
| RFS > 3 | 14 | 348 | 5 | 728 | P = 0.46 | 14 | 245 | 5 | 868 | P = 0.24 |
| P-value | **P = 0.0015** | | P: NC* | | | **P = 0.008** | | P: NC* | | |
| Stage 4 | | | | | | | | | | |
| ER positive | 6 | NR | 3 | 91 | P = 0.080 | 6 | NR | 3 | 109 | **P = 0.045** |
| ER negative | 12 | 348 | 2 | 728 | P = 0.54 | 12 | 348 | 2 | 868 | P = 0.10 |
| P-value | P = 0.16 | | P = 0.5 | | | **P = 0.015** | | P = 0.50 | | |

Subgroup analysis of E2 effect according to grade (1, 2, 3), stage (lymph node status/stage 4), bioscore (RFS) grouping, ER IHC (positive vs negative) in spayed versus intact dogs. Outcomes reported as median days. Endpoints: time to primary metastasis (TTM) and overall survival (OS), Log rank testing. NR: median not reached. *P: NC: not calculated

positive tumors (p = 0.045, Fig 9). The difference did not reach significance for TTM for this subgroup (p = 0.080, Table 4).

**Association between E2, tumor grade and ER expression.** Comparing the distribution of grade and ER expression between the high versus low E2 categories, revealed a relatively

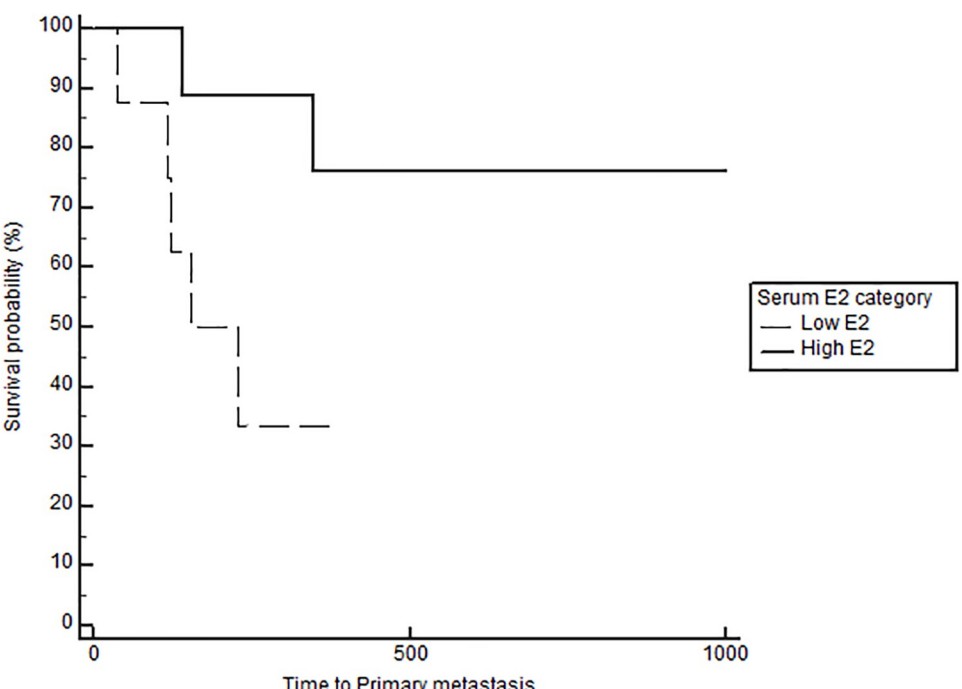

**Fig 4. Time to metastasis in <u>spayed</u> dogs with stage 4 disease according to serum E2 category, p = 0.047.**

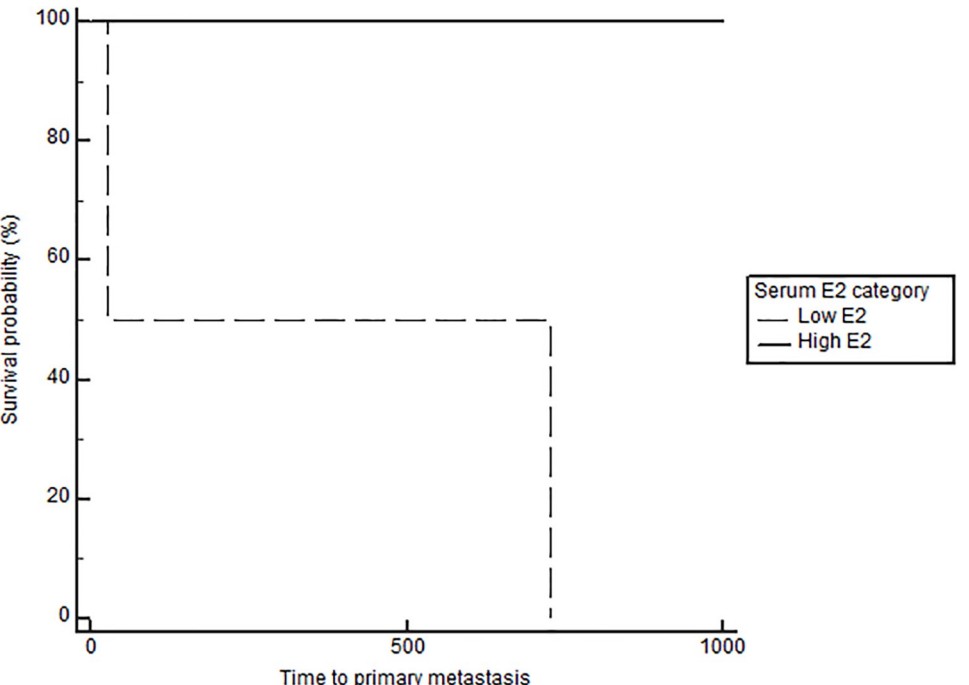

**Fig 5. Time to metastasis in <u>intact</u> dogs with stage 4 disease according to serum E2 category, p = 0.039.**

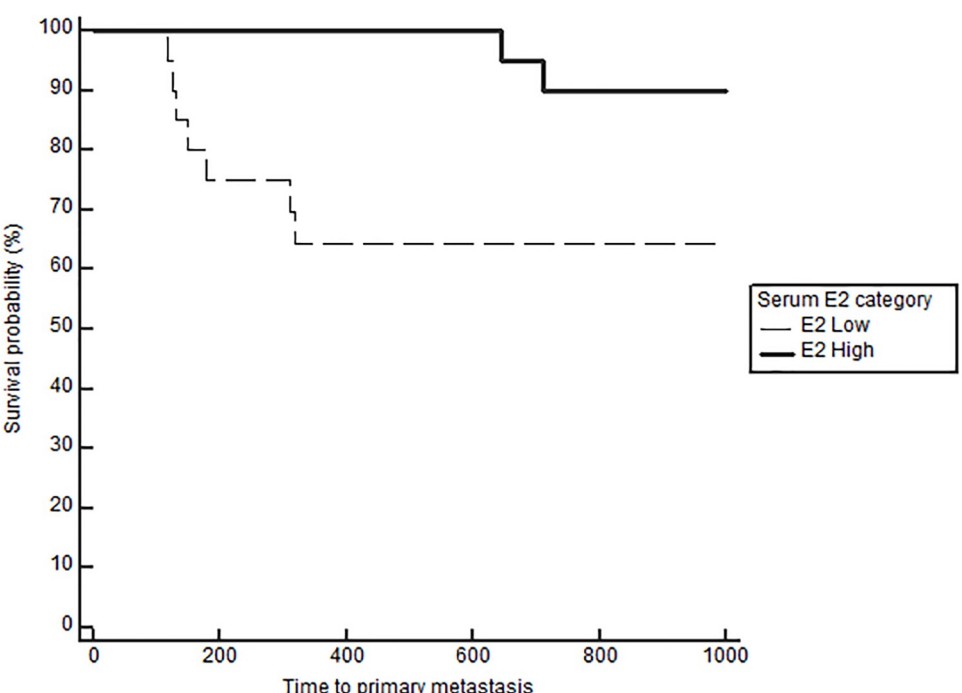

**Fig 6. Time to metastasis according to serum E2 category in dogs with <u>ER-positive</u> tumors that were <u>spayed</u>, p = 0.0052.**

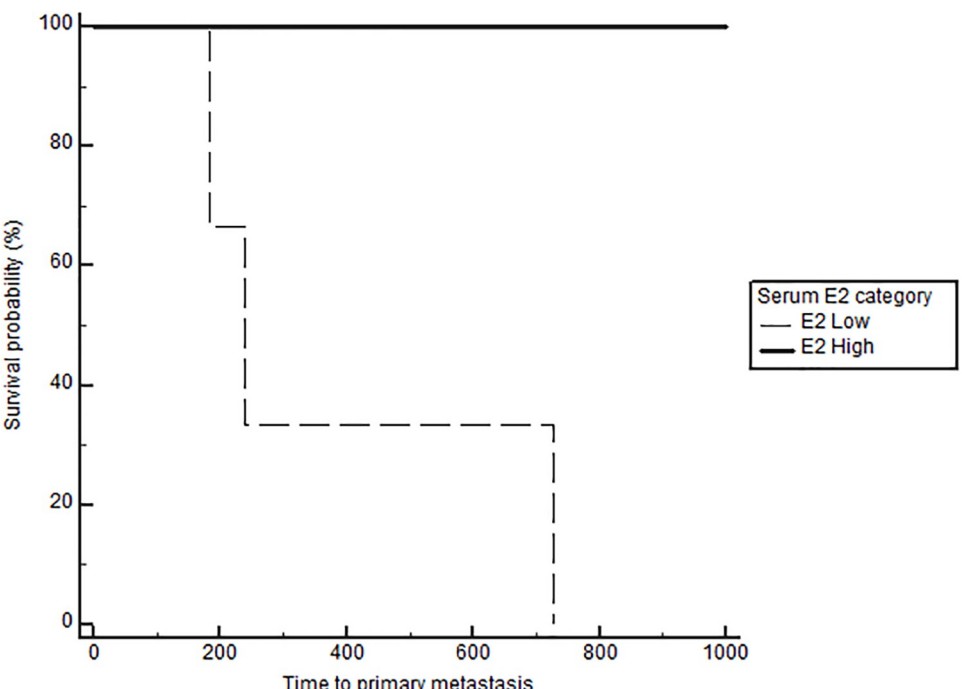

**Fig 7. Time to metastasis according to serum E2 category in dogs with <u>ER-negative</u> tumors that remained <u>intact</u>, p = 0.036.**

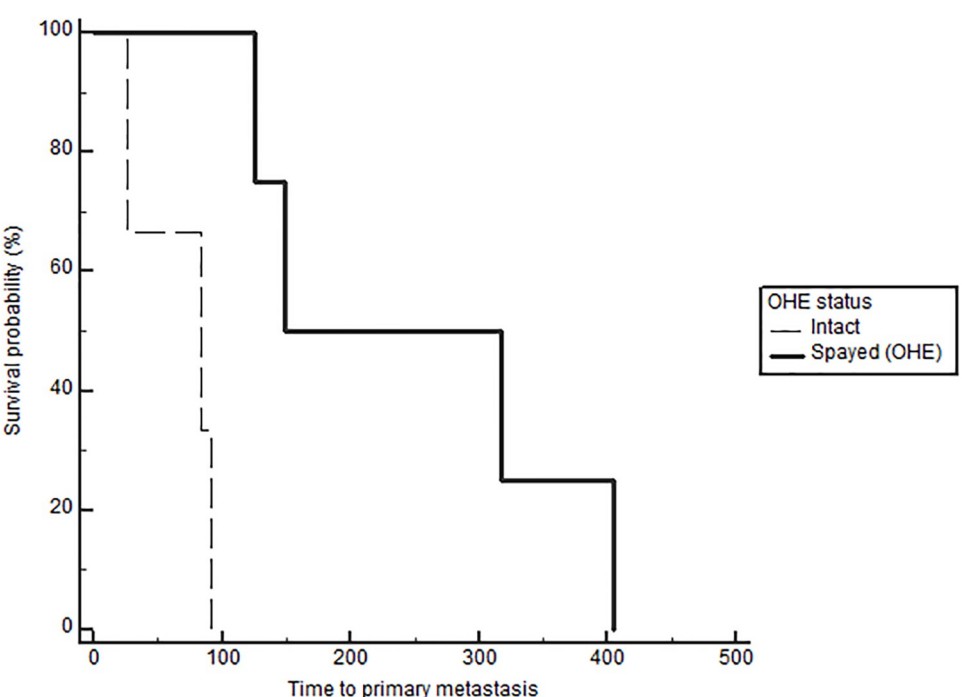

**Fig 8. Time to metastasis according to OHE status in dogs with high bioscore (RFS>3) and ER-positive tumors p = 0.0042.**

lower proportion of dogs with grade 3 tumors; only 5 of 15 dogs (33.3%) had high E2, compared to 39 of 67 dogs (58.2%) with grade 1 tumors, where the majority were in the high E2 category. This difference, however, was not statistically significant when E2 was analyzed as a categorical variable (Table 5). Similarly, 60.8% of ER positive tumors were in the high E2 category compared to 45% of the dogs with ER negative tumors. This was also not significant, according to chi square, Table 5. However, when analyzing the association between E2 and grade using linear regression where E2 was considered as a continuous variable, there was a significant inverse relationship (decreasing serum E2 with increasing tumor grade) between serum estrogen and tumor grade, p = 0.034, Fig 10. Similarly, when analyzing serum E2 and ER expression using linear regression there was a positive and statistically significant association between increasing serum E2 and ER positivity, p = 0.028, Fig 11.

**Effect of E2 on NMT and types of NMTs.** The median time to NMT (TNMT) in the spayed group was 549 days. There was no difference in TNMT development between dogs with low versus high E2, with a median TNMT of 533 versus 648 days, in the low versus high E2 group respectively, p = 0.69. However, when comparing serum estrogen between dogs with NMTs and dogs that had no tumor related failures during their follow-up period (no mammary tumor metastasis or other NMTs), significantly more dogs with low serum E2 developed NMTs compared to dogs with high E2 (p = 0.012, Table 5). Several types of non-mammary tumors developed in dogs in the post-operative period. NMTs in spayed dogs included: splenic hemangiosarcoma (n = 6, 5 of which were biopsy confirmed, and one was a tentative diagnosis based on based on clinical exam, imaging, and breed), adrenocortical carcinoma (n = 3), mast cell tumor (n = 2), lymphoma (n = 2), pheochromocytoma (n = 1), paraganglioma (n = 1), osteosarcoma (n = 3), liposarcoma (n = 1), cecal sarcoma (n = 1), perivascular wall tumor (n = 2), other soft tissue sarcoma (n = 1), oral melanoma (n = 1), pancreatic carcinoma (n = 1), cholangiocellular carcinoma (n = 1), colonic carcinoma (n = 1), and unspecified brain tumor

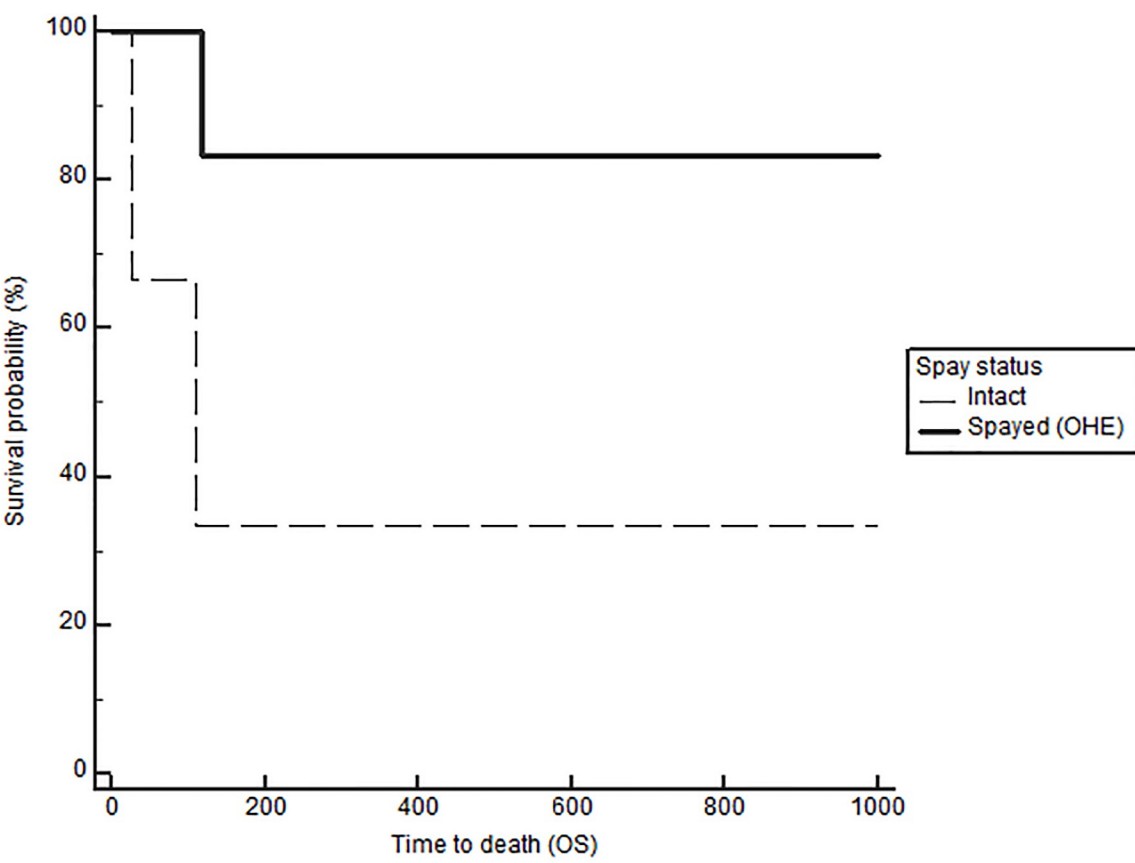

**Fig 9. Overall survival according to OHE status in dogs with LN-positive and ER-positive tumors, p = 0.045.**

Table 5. Association between E2, tumor grade, ER, and non-mammary tumors (NMT).

| Variable | n (%) | E2 category n high(H) vs low (L) | |
| --- | --- | --- | --- |
| Grade | | H | L |
| 1 | 67(56.3) | 39 | 28 |
| 2 | 37(31.1) | 15 | 22 |
| 3 | 15(12.6) | 5 | 10 |
| P-value | | p = 0.091 | |
| ER | | H | L |
| Positive | 51(56) | 31 | 20 |
| Negative | 40(44) | 18 | 22 |
| P-value | | P = 0.145 | |
| Tumor events | | H | L |
| New NMT Control[a] | 26(26) | 8 | 18 |
| P-value | 74(74) | 45 | 29 |
| | | P = 0.012 | |

Association between estrogen (E2, high versus low) and tumor grade, ER expression, development of non-mammary tumors (NMTs) in spayed dogs (Chi square).

[a] Control group are dogs with no tumor related events, including mammary tumor metastasis or non-mammary tumors)

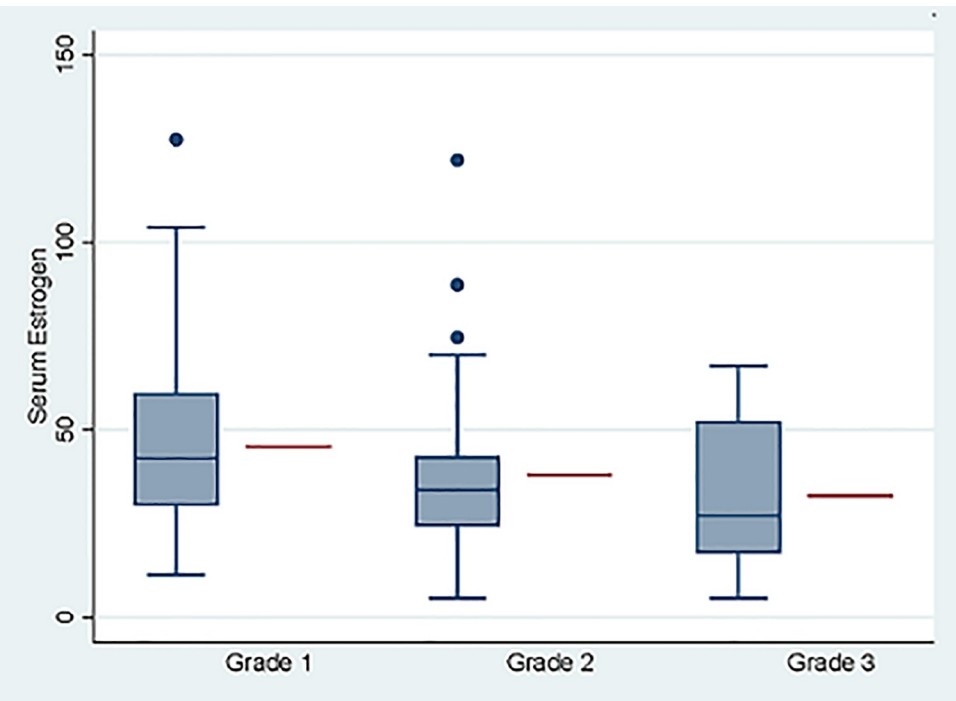

**Fig 10. Association between E2and tumor grade.** Serum estrogen (E2) analyzed as a continuous variable compared to tumor grade (grades 1, 2, 3), linear regression, p = 0.034. Blue boxes, whiskers and dots represent observed data and red line represents model adjusted mean for the respective group.

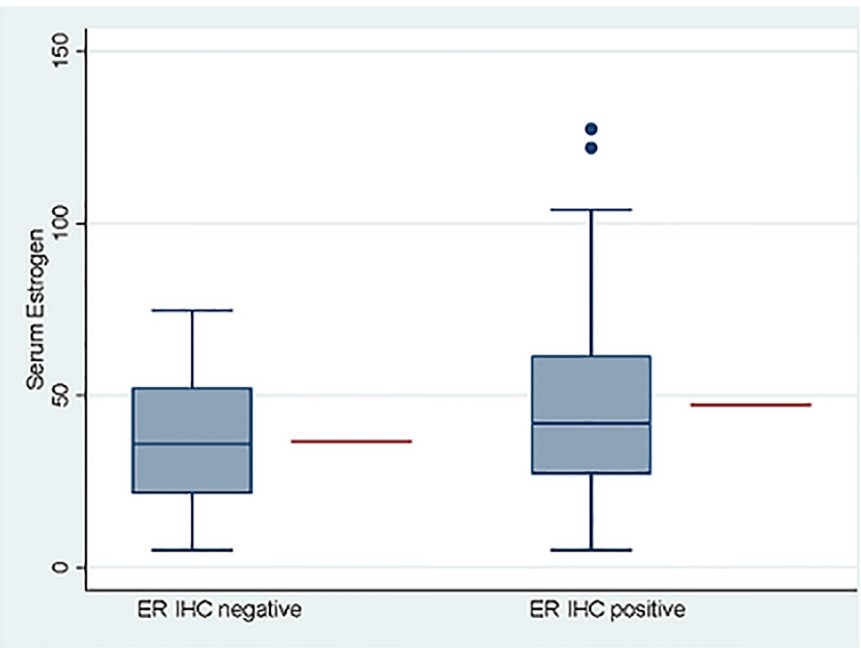

**Fig 11. Association between serum estrogen and tumor ER IHC (negative vs positive).** Serum estrogen (E2) analyzed as a continuous variable and compared to tumor ER IHC, linear regression, p = 0.028. Blue boxes, whiskers and dots represent observed data and red line represents model adjusted mean for the respective group.

**Table 6. Association between E2 and other tumor types.**

| Tumor category | n | E2 median pg/ml | Pairwise Comparison with Control |
|---|---|---|---|
| Hemangiosarcoma | 6 | 24.3 | P<0.0001 |
| Epithelial | 8 | 26.8 | P<0.0001 |
| Other Sarcomas | 8 | 33.8 | P = 0.239 |
| Hematopoietic | 4 | 58 | P = 0.516 |
| Miscellaneous | 2 | 40 | P = 0.551 |
| No NMGT or M1 (control group) | 74 | 42.4 | Control |

Post-hoc linear regression pairwise comparison of serum E2 in dogs with different types of postsurgical non-mammary tumors (NMT) and dogs without any tumor related post-surgical events (no mammary tumor metastasis or other NMTs) representing the control group.

(n = 1). Two dogs developed 2 separate malignancies during the post-operative period. Seven dogs developed both mammary carcinoma metastasis and a new NMT. Serum E2 was available for analysis in 5/6 dogs with post mammary tumor hemangiosarcomas of the spleen, all of which were in low E2 category with a mean value of 24.3 pg/ml. The NMTs were grouped in categories according to tissue of origin (epithelial, mesenchymal, hematopoietic) and their respective groups' median E2 value were compared to dogs with no tumor related outcomes (no M1 of NMGT). Hemangiosarcomas formed a group by themselves (Table 6). Three of the twenty-nine dogs in the intact group (10.3%) developed other non-mammary tumors (nasal tumor, soft tissue sarcoma, hepatocellular carcinoma).

## Discussion and conclusions

Estrogen has dual and opposing effects in both human and canine health. The positive impact associated with estrogen in women includes cardiovascular, skeletal, neurological, metabolic, reproductive and immunological health benefits [33–36]. The health benefits in canine health is mostly appreciated by studying the effects resulting from decreased estrogen via early ovariohysterectomy, such as an increased risk of urinary incontinence, degenerative orthopedic diseases such as hip dysplasia, cranial cruciate ligament rupture, as well as increased risk of various non-estrogen receptor expressing cancers including hemangiosarcoma, osteosarcoma, lymphoma, and mast cell tumors [37–44]. The increased risk of breast cancer or mammary tumors associated with estrogen exposure, however, is well recognized and indisputable in both species. Therefore, the results of this study may seem contradictory, if not implausible. Notably, there appears to be an overall benefit associated with high serum estrogen levels regardless of grade, stage and histological types. This is in part due to uneven distribution of grade between the high and low estrogen categories: Dogs with low grade tumors had higher serum estrogen levels than dogs with than dogs with grade 3 tumors (Fig 10) and this likely contributes to significantly longer survival in this group. Further subgroup analysis confirmed that serum estrogen level did not change the outcomes in dogs with low-grade tumors, but few dogs in this category developed primary metastasis. Most surprisingly, and consistent with results from our pilot study, is that high serum estrogen significantly improved the outcome in dogs with high-risk tumors, including advanced stage and high RFS bioscore. These tumors are often ER negative and therefore the pro-carcinogenic effect of estrogen via the receptor is not in play [31, 45, 46]. In this group, high serum estrogen appears to delay or prevent the development of mammary tumor metastasis. These results may suggest that removing the ovaries and therefore eliminating an important source of serum estrogen may be harmful to this

subset of dogs with high serum E2 and ER negative tumors (Tables 2, 3 and 4), and this finding was consistent in both the Cox regression analysis and subgroup analysis. However, this observation is based on a small subset of cases and thus susceptible to bias, but the effect on preventing metastasis is profound in a group of dogs with otherwise aggressive tumors, suggesting that E2 may have a uniquely powerful influence. Follow-up studies are clearly indicated to further examine this result.

A similar benefit is also seen in dogs with high serum E2 and ER positive tumors, but only in the spayed group, suggesting that lowering E2 via OHE may also prevent or slow progression in this particular subset of hormone dependent tumors (Tables 2, 3 and 4, Fig 6). Based on both subgroup analysis and Cox regression, dogs with high serum E2 and ER-positive tumors that underwent OHE have superior outcomes compared to all other categories. The conflicting effects of OHE in dogs with ER-positive/negative tumors support a dual role of E2; one as a driver of breast carcinogenesis via the ER, and the other as a cancer inhibitor, likely mediated via non-receptor dependent mechanisms. This may explain the opposite effects of hormonal deprivation via OHE depending on serum E2 group and tumor ER and underscores the importance of performing specific subgroup analysis to explore the complex interactions between serum E2, tumor ER expression, and OHE in dogs with mammary tumors. This effect may be obfuscated by the multivariable analysis.

Intact dogs developed significantly more new mammary tumors than dogs in the OHE group (58.6% versus 31.5%). These results are consistent with our previous study in dogs with benign mammary tumors and confirm that continued estrogen exposure after tumor removal contributes to increased risk of new mammary tumors and corroborates a pro-carcinogenic effect of estrogen [47]. Overall, there was no difference in the incidence of metastasis (M1) between intact versus spayed dogs even when stratifying for ER positive tumors. However, when evaluating the effect of OHE in a high-risk subgroup (RFS >3) and ER positive tumors, there was a significant improvement in metastasis free survival in dogs undergoing OHE, supporting the role of ovarian hormones in driving mammary carcinoma tumor progression via the receptor, and the similarities to human breast cancer. Additionally, dogs with lymph node positive, ER-positive tumors had significant improvements in OS if they were spayed compared to the intact dogs, similar to what has been documented in women with comparable, stage and tumor receptor status.

This potential protective role of estrogen in dogs with high risk/ER negative mammary tumors may also explain the lower incidence of later NMTs in the high serum E2 group and shed light on underlying mechanisms associated with the increased risk of various non-mammary cancers in dogs spayed at an early age as reported in several recent epidemiological studies [40–44]. It is worth noting that 6 dogs developed splenic hemangiosarcomas. All of the dogs (5/6) with serum E2 results available fell in the low serum E2 category. An increased risk of hemangiosarcoma has been reported in several epidemiological studies on the long-term health consequences of early OHE in female dogs. Dogs that developed epithelial malignancies after mammary tumor treatment had also significantly lower serum E2 than controls. In contrast to the epidemiological studies on the increased risk for lymphoma in dogs that are spayed at a young age, we found no difference between serum E2 in dogs with post mammary tumor sarcomas or hematopoietic malignancies and controls. Few dogs, however, were diagnosed with these subtypes, possibly because the majority was intact up until their mammary tumor surgery.

This phenomenon may not be unique to dogs. The role of estrogen in women parallels our data in terms of risk for breast cancer relapse and development of other cancers, particularly carcinomas. The use of soy supplements in women with a diagnosis of breast cancer is controversial and the data regarding benefit versus harm is conflicting. However, a recent large

prospective study found that women with ER negative breast cancer who took plant-estrogen containing supplements after surgery had a 20% improvement in all cause survival; this benefit was not observed in women with ER-positive tumors [48]. These results highlight the importance of analyzing outcomes based on the patients' tumor ER expression, and the similarities with our observations in dogs with ER-negative tumors. There are relatively few studies in women with breast cancer that have analyzed serum estrogen levels, and even fewer where the serum samples have been collected at a specific time point in their breast cancer diagnosis/treatment. This makes comparisons difficult, as some of the anti-estrogen treatments, including tamoxifen, will increase serum estrogen in pre-menopausal women, while aromatase inhibitors will decrease serum estrogen in postmenopausal women [49–51]. Despite these challenges, one study documented better outcomes in women with high serum estrone, but not estradiol, if they had ER-negative cancers. Notably, the serum samples were collected on average 30 months after the initial breast cancer diagnosis in these women, thus the effect of hormonal therapy might have influenced the results [52]. The complex effect of estrogen in human breast cancer is however, well known. Several decades ago, estrogen was often used as rescue therapy in women with advanced breast cancer; one study reported a response rate of 65% with estrogen therapy compared to 76% with oophorectomy in women with ER positive tumors. Interestingly, the response rate was only 10% in women with ER negative tumors [21, 53–55]. Later research has confirmed these findings and provided mechanistic explanation for this apparent paradoxical effect by showing that estrogen can induce apoptosis via binding to the ER in cancer cells that have become resistant to traditional hormonal therapy [56, 57]. These results illustrate the plasticity of the breast cancer cells and the duplicitous role of estrogen in this malignancy.

Today, the use of estrogen to treat breast cancer has become very controversial in light of the awareness of estrogens' role in breast carcinogenesis. Nevertheless, there is significant evidence in the research literature that estrogen has general health benefits and anti-cancer effects: Women who loose ovarian function at an early age have a significantly increased risk of dying prematurely from any cause, including coronary heart disease or cancer [58]. A large prospective observational study evaluated the long-term outcomes in women undergoing complete or partial ovarian resection and confirmed the increased all-cause mortality rate in women who underwent total oophorectomy prior to age 50. Further subgroup analysis also showed that women who did not take hormone replacement therapy (HRT) after total oophorectomy had a significant increased risk of lung cancer compared to women who retained some ovarian tissue [59]. These findings are complimented by other studies showing that HRT also significantly reduces lung cancer risk [60–62]. A recent meta-analysis found a 30% increased incidence of colorectal cancer in women who underwent oophorectomy [63]. As in lung cancer, these results are supported by an earlier study that showed a significant decreased incidence of colon cancer in women taking HRT [64]. These examples of increased incidence of certain cancers associated with premature depletion of natural estrogen and the protective role of HRT, may support the findings in our study. The overall risk for NMTs in our study was relatively low (10 and 20%), and there was no difference between the spayed and the intact groups. This may in part be due to the fact that these dogs were intact until they developed mammary tumors as middle-aged to older dogs. The beneficial effect of hormones is cumulative and likely sustained, even after OHE, as illustrated by the fact that the median time to other non-mammary tumors is 560 days.

Another interesting finding in this research is the association between grade and serum estrogen. Specifically, dogs with low grade tumors had higher serum estrogen compared to dogs with grade 3 tumors. This suggests that estrogen may influence the natural biology of mammary tumors by modulating the genotype and subsequently influence the phenotype of

the carcinoma. It is possible that higher serum estrogen provides a selective growth advantage to ER positive epithelial cells, leading to a more differentiated and low-grade tumor. The association between ER positivity and high serum E2 may support this theory. Additionally, the presence of potential negative feedback mechanisms present in ER positive breast epithelial cells may counteract unrestricted cellular proliferation, which would otherwise lead to mutations and more aggressive sub-clones. Low serum estrogen may in fact "take the breaks" of this internal control and drive the cellular machinery towards adapting to a more hostile hormone-deprived environment, utilizing alternative non-estrogen responsive/dependent genes which ultimately may result in a more aggressive and hormone-independent phenotype. This negative feedback loop has been documented in healthy, non-cancerous mammary epithelial cells and is assumed a necessary internal control to prevent unrestricted cellular proliferation in breast tissue [65]. Further research to explore the effect of estrogen on gene expression in mammary carcinomas, specifically the difference between high- and low-grade tumors, will be necessary to understand this effect.

There are several limitations in this study: Data used in this study was collected from 2 separate prospective trials conducted on 2 different continents in client-owned and shelter dogs with naturally occurring mammary tumors. Both studies, however, were designed and executed by the same investigator who oversaw and/or conducted the majority of clinical exams and follow-up diagnostics. The pathologists involved in these 2 studies were all experienced in mammary tumor pathology, and a high percentage of cases underwent post-mortem exams to confirm the presence of metastasis and presence of other tumors. Most dogs in this study were spayed (n = 130) and only 29 dogs from the MAF study were randomized to remain intact. Ideally, we would like to have a more even distribution between intact and spayed dogs, however, this was a retrospective analysis of data from studies already completed and could not be changed. Because of this, our statistical power and P-values were sometimes borderline, and some subgroups were small, thus the findings might be skewed. However, our results from Cox-regression analysis and log rank testing confirm the same associations between hormonal factors and outcome. Nevertheless, a prospective randomized controlled trial will always be preferred over a retrospective analysis, therefore, further confirmatory studies are encouraged to confirm of refute these findings.

Furthermore, the assays for serum E2 and tumor ER IHC differed between the studies. Despite the methodological differences, the median serum estrogen values between the 2 groups were quite similar (35 vs 39 pg/ml) and did not differ statistically. Similarly, when combining dogs from both groups as either ER positive or negative based on ER IHC, we confirmed significant association with outcome, as well as the association between ER and OHE. This suggests that both IHC assays recognize the ER despite differences in primary antibodies and methods; providing mutual validation of the respective IHC assays and possibly strengthening our findings. There might be some, likely minor, differences in sensitivity between the 2 methods, but nevertheless the association with outcome was significant in the pooled analysis. Notably, there are numerous ER antibodies available and used for diagnostic testing in human breast cancer and these are often combined as positive versus negative whenever studies are included in meta-analysis. According to the American Society of Clinical Oncology and College of American Pathologists (ASCO/CAP) guidelines the antibodies selected should be validated through clinical testing against patients' outcome [66]. Here we show that the pooled ER staining results predict overall better outcomes (Fig 2), and is particularly important in high risk tumors because the results may inform therapy, similar to how decisions regarding hormonal therapy are made in human breast cancer. Specifically, dogs with RFS > 3 and ER positive tumors benefited from OHE, both in terms of time to metastasis and overall survival, see Table 4, Fig 8.

Translational and comparative oncology research is built on similarities between human and animal cancer models. In general, similarities strengthen these models; findings in one model can be applied directly to the other. Differences may complicate such research. The dissimilarities in estrus cycles between women and dogs can be considered such a complicating difference. However, it may be exactly these differences and the prolonged periods between each estrus that allow us to look at the interactions through a clearer lens. The biological and molecular effects of estrogen on their target receptors and tissues are likely the same, thus allowing us to study cancer in natural estrogen rich or estrogen deprived environment. Collectively, these results reflect the complexity of the role of estrogen in cancer and the plasticity of breast cancer cells and their ability to adapt to hostile conditions. Contrary to the accepted dogma, estrogen may have beneficial effects in a subset of breast cancer patients, and cancer more broadly. The spontaneous dog model of mammary carcinoma and the differences in estrus cycles provides a unique resource in breast cancer research and may contribute to improved understanding of the complex effects of estrogen to better harness the prognostic and therapeutic possibilities that estrogen manipulation may offer.

## Supporting information

**S1 Table. Histological and clinical prognostic factors in dogs with mammary carcinomas.** Prognostic variables in 159 dogs with mammary carcinomas. Univariate and multivariable Cox regression analysis. Variables: tumor size ($<$3,3–5,$>$5), Tumor grade (1, 2, 3) Histological subtypes: 1: carcinomas, 2: complex carcinomas, 3: carcinoma arising in benign mixed tumor, 4: other types including solid carcinomas, comedocarcinoma, carcinoma and malignant myoepithelioma, anaplastic carcinoma, carcinosarcoma, Who stages (1–4), Presence/absence of vascular invasion, and surgical margins: clean vs incomplete. Endpoint: time to primary metastasis. Hormonal factors (serum estrogen, tumor estrogen receptor and spay status not included. This dataset consist of pooled data from previously published studies (28, 29). *Hazard Ratio.
(DOCX)

## Author Contributions

**Conceptualization:** Karin U. Sorenmo.

**Data curation:** Karin U. Sorenmo, Veronica Kristiansen.

**Formal analysis:** Karin U. Sorenmo, Amy C. Durham, Enrico Radaelli, Darko Stefanovski.

**Funding acquisition:** Karin U. Sorenmo.

**Investigation:** Karin U. Sorenmo, Amy C. Durham, Enrico Radaelli, Veronica Kristiansen.

**Methodology:** Karin U. Sorenmo, Amy C. Durham, Enrico Radaelli, Laura Peña, Michael H. Goldschmidt.

**Project administration:** Karin U. Sorenmo.

**Resources:** Karin U. Sorenmo, Amy C. Durham.

**Software:** Darko Stefanovski.

**Supervision:** Karin U. Sorenmo, Amy C. Durham.

**Validation:** Karin U. Sorenmo, Amy C. Durham, Enrico Radaelli.

**Visualization:** Amy C. Durham, Enrico Radaelli, Laura Peña, Michael H. Goldschmidt.

**Writing – original draft:** Karin U. Sorenmo.

**Writing – review & editing:** Karin U. Sorenmo, Amy C. Durham, Enrico Radaelli, Veronica Kristiansen, Laura Peña, Darko Stefanovski.

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
