## [Decision Letter · Decision Letter 0]

28 Aug 2019

[EXSCINDED]

PONE-D-19-21030

The Estrogen effect; clinical and histopathological evidence of dichotomous influences in dogs with spontaneous mammary carcinomas.

PLOS ONE

Dear Dr. Sorenmo,

Thank you for submitting your manuscript to PLOS ONE. After careful consideration, we feel that it has merit but does not fully meet PLOS ONE’s publication criteria as it currently stands. Therefore, we invite you to submit a revised version of the manuscript that addresses the points raised during the review process.

Please address all Reviewer comments. I specifically agree with Reviewer 1 regarding the need to incorporate additional variables for potential prognostic significance, including histologic subtype, histologic grade, WHO/TNM stage, surgical margins, presence/absence of lymphatic/vascular invasion, and use of any adjuvant therapies.

We would appreciate receiving your revised manuscript by Oct 12 2019 11:59PM. To enhance the reproducibility of your results, we recommend that if applicable you deposit your laboratory protocols in protocols.io, where a protocol can be assigned its own identifier (DOI) such that it can be cited independently in the future. For instructions see: http://journals.plos.org/plosone/s/submission-guidelines#loc-laboratory-protocols

We look forward to receiving your revised manuscript.

Kind regards,

Douglas H. Thamm, V.M.D.

Academic Editor

PLOS ONE

Journal Requirements:

2. In your Methods section, please provide additional details regarding participant consent from the animals owners. In the ethics statement in the Methods and online submission information, please ensure that you have specified (1) whether consent was informed and (2) what type you obtained (for instance, written or verbal). If the need for consent was waived by the ethics committee, please include this information.

4. Please ensure that you refer to Figure 6 in your text as, if accepted, production will need this reference to link the reader to the figure.

Reviewers' comments:

Reviewer's Responses to Questions

**Comments to the Author**

1. Is the manuscript technically sound, and do the data support the conclusions?

Reviewer #1: Partly

Reviewer #2: Yes

2. Has the statistical analysis been performed appropriately and rigorously? 

Reviewer #1: Yes

Reviewer #2: No

3. Have the authors made all data underlying the findings in their manuscript fully available?

Reviewer #1: Yes

Reviewer #2: Yes

4. Is the manuscript presented in an intelligible fashion and written in standard English?

Reviewer #1: No

Reviewer #2: Yes

5. Review Comments to the Author

Reviewer #1: The authors were suggested that the benefit from high serum estrogen would be strongest in dogs with histologically aggressive tumors, ER-negative tumors, and high bioscores – all indicators of a worse prognosis, possibly by activating the non-estrogen-receptor pathway. Thus, high serum estrogen would be significant in the survival benefit in dogs. The current case has novelty. However, there are several minor issues to be corrected in the manuscript.

- The quality of English redaction should be improved.

- Would like to encourage the authors to analyze expression levels both mRNA and protein level by Real-Time PCR and Western Blott methods respectively if it is possible. This makes your experimental data stronger and more accurate. Principally, regarding the suggestion of the other pathway of action the estrogen in serum.

- No pattern in analysis: Different techniques for serum estrogen analysis and different antibody for immunohistochemistry, made in different laboratory.

- Low number of intact dog (29) regarding neutered dogs (130) – there are a large heterogeneity between group.

- Some questions need to be answered, because they are so relevant to this paper methodology:

* Which is the primary neoplasm histologic type? All neoplasms cannot be grouped and analyzed together. It’s known that they are different behaviors. Complex carcinomas never will be the same clinical behavior that carcinosarcoma or micropapillary carcinomas.

* Absence of treatment approach: Surgery (which approach?), spayed time (1st, 2nd, 3rd or latted?), radiotherapy or adjuvant chemotherapy approach?

How to know whether the estrogen or these others clinical information influenced the new non mammary tumors?

- The discussion is the most important part of the paper. It is at this moment, the authors should interpret and describe the significance of your findings in light of what was already known about the research problem being investigated, and to explain any new understanding or insights about the problem after you've taken the findings into consideration. So this part should be improved by a more comprehensive manner and written by using more fluent language.

- The study limitations were also not mentioned enough in the discussion.

Reviewer #2: I thank the authors for an interesting and thoughtful manuscript. As a veterinarian I should be careful to criticize a study containing prospective data, as this is still rare in our field. However, the samr type of caution must be for retrospective as well as prospective data. Hence, I would like to share some thoughts that may improve the presentation of data?

General:

I would strongly recommend the authors to not use borderline significance, unless more stringent discussion on how to interpret the data should be made. This more specific is regarding the longer time to metastasis in dogs with positive ER expression only in the spayed group (p=0.049). This as the groups are so different in size (130 spayed and 29 intact). This will risk a bias and over interpretation of result.

I think the authors are doing a good job in trying to describe the paradoxal dual effect of Estrogen on both being tumor promoting and tumor protectant, depending on tumor type and time of exposure to Estrogen.

Figures

In my copy the intact and dashed lines are not clear enough and the interpretation of the survival curves are not clear enough. I recommend you to make this clearer.

6. PLOS authors have the option to publish the peer review history of their article (what does this mean?). If published, this will include your full peer review and any attached files.

Reviewer #1: No

Reviewer #2: Yes: Henrik Ronnberg

---

## [Author Response · Author response to Decision Letter 0]

16 Sep 2019

All comments from editor and reviewers have been addressed in the file called "Response to reviewers"

Response to reviewers and editor:

Reviewer #1: The authors were suggested that the benefit from high serum estrogen would be strongest in dogs with histologically aggressive tumors, ER-negative tumors, and high bioscores – all indicators of a worse prognosis, possibly by activating the non-estrogen-receptor pathway. Thus, high serum estrogen would be significant in the survival benefit in dogs. The current case has novelty. However, there are several minor issues to be corrected in the manuscript.

- The quality of English redaction should be improved.

Response from Authors: We have done another very careful edit of the manuscript and very few mistakes were noted. Please provide a specific example of where you think the English needs improvement. 

- Would like to encourage the authors to analyze expression levels both mRNA and protein level by Real-Time PCR and Western Blott methods respectively if it is possible. This makes your experimental data stronger and more accurate. Principally, regarding the suggestion of the other pathway of action the estrogen in serum.

Response from authors: We agree that mRNA, PCR and western Blot analysis would be interesting and informative. But this is a completely different study and research questions. Here we are analyzing clinical, histological and hormonal factors, their interactions and association with outcome. We hope that we can continue this work and utilize the additional analysis as suggested or that other researchers will see this work and become intrigued over the associations. However, these suggestions are beyond the purpose of this work.

- No pattern in analysis: Different techniques for serum estrogen analysis and different antibody for immunohistochemistry, made in different laboratory.

Response from authors: Yes, both the E2 and ER IHC were analyzed by different laboratories because we used 2 different sources (MAF dogs and Shelter dogs). However, despite different methodology in E2 analysis, the results were almost identical (as illustrated by the comparison of values included in the manuscript), thus one might propose that these different methods even though they are from different laboratories and different continents provide each other mutual validation. 

Similarly, we included the use of 2 different ER IHC antibody, technique and laboratory. Both methods included positive and negative controls and utilized commercially available antibodies which both were stated to recognize ER in dogs. Please note that in women numerous different antibodies are used to detect ER for therapeutic and research purposes. There might be slight differences in sensitivities between the antibodies and the ability to detect low expression, but nevertheless when pooling the 2 groups ER maintained a very strong prognostic significance together. These results reflect the prognostic importance of ER, and the association with E2 and OHE also confirms its biological and clinical importance. We would have preferred to have all samples stained with the same ER and technique but that was not possible here, and similar to the situation with E2, using 2 different antibodies and still maintaining the associations may strengthen our findings. 

We have inserted a new paragraph in the discussion where this has been addressed and we have also included another reference at the end of the paragraph to support our position.

- Low number of intact dog (29) regarding neutered dogs (130) – there are a large heterogeneity between group.

Response from Authors:

We used data collected from 2 separate studies, the MAF study where 31 dogs were spayed and 29 remained intact and shelter dogs that participated in the PennVet Shelter Canine mammary tumor Program. All these dogs had to be spayed as part of their treatment as per shelter policies. Thus, the ratio between intact and spayed dogs is skewed and cannot be changed since we were utilizing data from 2 studies that were already completed. The study design is best described as a cross-sectional retrospective study and clearly is an inferior study design compared to a prospective randomized controlled trial. However, the data is prospectively collected for each study and represents high quality clinical data with excellent outcome information, therefore, statistical comparisons are possible, and the results should not be dismissed. 

This has also been addressed in the new paragraph in the discussion where we address the 3 points regarding E2, ER and the uneven distribution between intact and spayed dogs 

- Some questions need to be answered, because they are so relevant to this paper methodology:

* Which is the primary neoplasm histologic type? All neoplasms cannot be grouped and analyzed together. It’s known that they are different behaviors. Complex carcinomas never will be the same clinical behavior that carcinosarcoma or micropapillary carcinomas.

Response to authors: 

We agree that canine mammary tumors represent a diverse group of tumors. We would like to bring the reviewer’s attention to our previous publications, 28 and 29 which are based on the same data sets and details exactly this diversity in behavior. These manuscripts provide detailed information regarding staging, surgical treatment and follow-up. We have added the details requested in this new manuscript. However, the purpose of this manuscript was to study the association between hormonal factors, histological factors and outcome to show the diversity in outcomes and identify where the hormonal factors play a role. Please note that we have analyzed the association between hormonal factors and the refined flexible score (RFS), see table 4. This score is based on tumor size, grade, histological subtype and lymph node status. Thus, the data regarding the primary tumor type is included in the analysis as it contributes to the bioscore (ref 29). As you will see, table 4 also contains detailed data regarding the association between grade, lymph node status and ER/E2. 

Many of the prognostic variables in canine mammary tumors are closely associated with each other, as shown in previous publications. According to our previous publication, only grade and tumor size remained significant in the multivariate analysis where of the prognostic factors (stage, grade, subtypes, vascular invasion, and margins) were included, ref 29. 

However, as a response to the request from reviewer 1 and the associated editor, we have included a supplemental table (S1) where all these individual factors (Tumor size, grade, histological type, WHO stage, presence of vascular invasion and surgical margins are provided) without including the effect of hormonal factors since this is already provided in table 4. Here we provide an analysis of both data sets together (MAF and Shelter dogs) and the sum of a pooled analysis which has not been published previously. Please see the supplemental table 1 (S1)

* Absence of treatment approach: Surgery (which approach?), spayed time (1st, 2nd, 3rd or latted?), radiotherapy or adjuvant chemotherapy approach?

Response from authors: This has been added to the manuscript. Specifically: Only 2 dogs received chemotherapy, both were spayed as part of their treatment, one with a grade 3 tumor which developed metastasis and died from her disease, the other dog with grade 1, stage 4 disease was alive without relapse 944 days after surgery. Please note that none of the dogs (n=6) with grade 1, stage 4 disease developed metastatic disease during follow-up and only one of these received chemotherapy (ref 29). 

Radiotherapy does not have an established role in the treatment of canine mammary carcinomas (other that possibly for palliative intent)

How to know whether the estrogen or these others clinical information influenced the new non mammary tumors?

Response from reviewers: We believe we have addressed and discussed these associations in the discussion. We have provided extensive epidemiological data from both veterinary medicine and human medicine to explain and support our findings and support the hypothesis that estrogen provide general anti-cancer activity in non-ER tumors. Further research to confirm the specific pathways and elucidate the mechanisms is indicated based on our results. 

- The discussion is the most important part of the paper. It is at this moment, the authors should interpret and describe the significance of your findings in light of what was already known about the research problem being investigated, and to explain any new understanding or insights about the problem after you've taken the findings into consideration. So this part should be improved by a more comprehensive manner and written by using more fluent language.

Response from reviewers: Please see above, we believe we have incorporated all the important and sometimes contradictory findings as it relates to the estrogen effect and established a plausible narrative where we discuss the finings and how they are supported by previous epidemiological and biological research relating to estrogen and the estrogen receptor. We appreciate the positive comments from reviewer 2 regarding the discussion. 

- The study limitations were also not mentioned enough in the discussion.

We have included a new paragraph in the discussion where all the limitations are summarized. Please see the revised manuscript

Reviewer #2: I thank the authors for an interesting and thoughtful manuscript. As a veterinarian I should be careful to criticize a study containing prospective data, as this is still rare in our field. However, the samr type of caution must be for retrospective as well as prospective data. Hence, I would like to share some thoughts that may improve the presentation of data?

General:

I would strongly recommend the authors to not use borderline significance, unless more stringent discussion on how to interpret the data should be made. This more specific is regarding the longer time to metastasis in dogs with positive ER expression only in the spayed group (p=0.049). This as the groups are so different in size (130 spayed and 29 intact). This will risk a bias and over interpretation of result.

Response from reviewers: We have included a new paragraph in the discussion where all the limitations and possible shortfalls including different size between groups that were spayed vs remained intact. We have also included a sentence specifically related to statistical power: “Most dogs in this study were spayed (n=130) and only 29 dogs from the MAF study were randomized to remain intact. Ideally, we would like to have a more even distribution between intact and spayed dogs, however, this was a retrospective analysis of data from studies already completed and could not be changed. Because of this, our statistical power and P-values were sometimes borderline, and the results might be skewed; therefore, further confirmatory prospective randomized studies are encouraged to confirm of refute these findings”.

I think the authors are doing a good job in trying to describe the paradoxal dual effect of Estrogen on both being tumor promoting and tumor protectant, depending on tumor type and time of exposure to Estrogen.

Response from authors: We appreciate it!

Figures

In my copy the intact and dashed lines are not clear enough and the interpretation of the survival curves are not clear enough. I recommend you make this clearer.

Response from Reviewers: We would be happy to redo the curves, but it is not clear whether this is only an issue for this particular reviewers’ copy. We received no comments from the technical screening of the manuscript regarding this. Please advise. 

Comments from Academic editor, Dr Douglas Thamm: I specifically agree with Reviewer 1 regarding the need to incorporate additional variables for potential prognostic significance, including histologic subtype, histologic grade, WHO/TNM stage, surgical margins, presence/absence of lymphatic/vascular invasion, and use of any adjuvant therapies.

Response from Authors: We have included a supplemental table 1 where all these individual prognostic variables and associated significance are provided for the whole group (n=159) without including the impact of hormonal factors (we assume this is what you requested). We would also like to point out that table 4 contains much of this information but in context of how these variables interact with hormonal factors. Specifically, we have evaluated the impact of serum estrogen on grade and stage (here stage 4) and ER IHC on RFS (which includes tumor size, grade and histological subtypes) and the effect of OHE. Similar evaluation is also performed to assess the impact of ER and OHE on LN positive tumors. We think these are the evaluations that illustrate the complex interactions between hormonal factors which was the main purpose of this manuscript. The Cox regression analysis presented in tables 2, 3 and 4 supports these results and strengthen our data from these subgroup analyses. 

Regarding the use of adjuvant therapies: We have included this information in the revised manuscript. As you will see, 2 dogs received FAC chemotherapy. These outcomes in these 2 dogs are the same as dogs that did not receive adjuvant chemotherapy. The dogs with stage 3, grade 3 tumor died due to metastatic disease and the dog with stage 4, grade 1 disease is alive 900+ days after surgery.

---

## [Decision Letter · Decision Letter 1]

16 Oct 2019

The Estrogen effect; clinical and histopathological evidence of dichotomous influences in dogs with spontaneous mammary carcinomas.

PONE-D-19-21030R1

Dear Dr. Sorenmo,

We are pleased to inform you that your manuscript has been judged scientifically suitable for publication and will be formally accepted for publication once it complies with all outstanding technical requirements.

With kind regards,

Douglas H. Thamm, V.M.D.

Academic Editor

PLOS ONE

Additional Editor Comments (optional):

Reviewers' comments:

Reviewer's Responses to Questions

**Comments to the Author**

1. If the authors have adequately addressed your comments raised in a previous round of review and you feel that this manuscript is now acceptable for publication, you may indicate that here to bypass the “Comments to the Author” section, enter your conflict of interest statement in the “Confidential to Editor” section, and submit your "Accept" recommendation.

Reviewer #1: All comments have been addressed

Reviewer #2: All comments have been addressed

2. Is the manuscript technically sound, and do the data support the conclusions?

Reviewer #1: Partly

Reviewer #2: Yes

3. Has the statistical analysis been performed appropriately and rigorously? 

Reviewer #1: No

Reviewer #2: Yes

4. Have the authors made all data underlying the findings in their manuscript fully available?

Reviewer #1: Yes

Reviewer #2: Yes

5. Is the manuscript presented in an intelligible fashion and written in standard English?

Reviewer #1: Yes

Reviewer #2: Yes

6. Review Comments to the Author

Reviewer #1: (No Response)

Reviewer #2: I thank the authors for addressing the points lifted and the quality of the manuscript has increased. I have no further questions.

7. PLOS authors have the option to publish the peer review history of their article (what does this mean?). If published, this will include your full peer review and any attached files.

Reviewer #1: No

Reviewer #2: Yes: Henrik Rönnberg

---

## [Editor Report · Acceptance letter]

18 Oct 2019

PONE-D-19-21030R1 

The Estrogen effect; clinical and histopathological evidence of dichotomous influences in dogs with spontaneous mammary carcinomas. 

Dear Dr. Sorenmo:

I am pleased to inform you that your manuscript has been deemed suitable for publication in PLOS ONE. Congratulations! Your manuscript is now with our production department. 

With kind regards,

on behalf of

Dr. Douglas H. Thamm 

Academic Editor

PLOS ONE